# FUNCTIONAL INTERPOLATION FOR RELATIVE POSITIONS IMPROVES LONG CONTEXT TRANSFORMERS

**Shanda Li**[1]***Chong You**[2], **Guru Guruganesh**[2], **Joshua Ainslie**[2], **Santiago Ontanon**[2]
**Manzil Zaheer**[3], **Sumit Sanghai**[2], **Yiming Yang**[1], **Sanjiv Kumar**[2], **Srinadh Bhojanapalli**[2]
[1]Carnegie Mellon University    [2]Google Research    [3]Google DeepMind
shandal@cs.cmu.edu

## ABSTRACT

Preventing the performance decay of Transformers on inputs longer than those used for training has been an important challenge in extending the context length of these models. Though the Transformer architecture has fundamentally no limits on the input sequence lengths it can process, the choice of position encoding used during training can limit the performance of these models on longer inputs. We propose a novel functional relative position encoding with progressive interpolation, FIRE, to improve Transformer generalization to longer contexts. We theoretically prove that this can represent some of the popular relative position encodings, such as T5's RPE, Alibi, and Kerple. We next empirically show that FIRE models have better generalization to longer contexts on both zero-shot language modeling and long text benchmarks.

## 1 INTRODUCTION

Transformer based Language Models have demonstrated state-of-the-art zero-shot performance on many natural language processing tasks (Brown et al., 2020), enabling increasingly longer context applications such as chat bots (Roller et al., 2021; Zhang et al., 2020b) and long document summarization and question answering (Zhang et al., 2020a; Guo et al., 2022; Ainslie et al., 2023). However, the accuracy of these models usually drops quickly for inputs longer than the ones used during training (Press et al., 2022; Anil et al., 2022; Deletang et al., 2023) – which are usually relatively short (e.g. 2048 for LLaMA (Touvron et al., 2023a;b)) to avoid the expensive quadratic attention cost during training. This has led to a significant interest in improving *length generalization* of Transformers - where we train the model using shorter inputs (e.g. 2048) and test the models performance on longer inputs (e.g. 8192) (Press et al., 2022; Anil et al., 2022; Chi et al., 2022; 2023; Chowdhury & Caragea, 2023; Chen et al., 2023).

Transformers are fundamentally permutation equivariant, and are agnostic to input sequence ordering (Vaswani et al., 2017; Yun et al., 2019)[1]. They rely on position encodings to learn the ordering of input tokens. Popular position encodings such as Absolute Positional Encoding (APE) (Vaswani et al., 2017) and more recent Rotary Positional Encoding (RoPE) (Su et al., 2021) do not generalize to longer contexts than seen during training (Kazemnejad et al., 2023). T5's relative positional encoding (Raffel et al., 2019) generalizes to longer contexts by using the same representation for all out of distribution (OOD) sequence lengths, but suffers from slow vector operations on modern accelerators (Press et al., 2022). Another line of recent work promotes length generalization by encoding specific inductive biases on how attention should decay with sequence length (Press et al., 2022; Chi et al., 2022; 2023). More recently, Kazemnejad et al. (2023) show that having no position encodings in decoder-only models can have better length generalization, albeit for small-scale synthetic tasks.

In this work we take a functional approach to learn the relative position biases[2], instead of having hard coded inductive biases, towards training language models with length generalization (focusing on decoder-only models). We propose **FIRE** (**F**unctional **I**nterpolation for **R**elative Positional **E**ncoding) method that i) uses a *learnable function* to map the input positions to biases, and ii) uses a *progressive*

---

*Work done during internship at Google Research.

[1]Note that decoder-only models can infer position from the causal attention mask (Haviv et al., 2022).

[2]We consider relative position encodings for their superior performance over absolute position encodings (Raffel et al., 2019; Chen et al., 2021).

*interpolation* technique, which ensures bounded input for the position encoding function for *all* input sequence lengths, thereby enabling length generalization.

A *functional* approach to learn the biases allows the model to adapt to the given task instead of always having the same inductive bias, e.g. bias towards nearby tokens as in (Press et al., 2022; Chi et al., 2022; 2023). In particular we use an MLP to learn these biases, which we theoretically *prove* can represent several popular methods such as T5's RPE, Alibi, and Kerple in a parameter efficient manner. In fact, all our experiments use a tiny MLP with a hidden size of 32, which is also accelerator-friendly unlike T5's RPE. Next, our *progressive interpolation* technique normalizes the query-key relative distance by the query position. Since for causal attention in language models the relative distance is always between 0 and the query position, progressive interpolation results in an output that is always bounded between $[0, 1]$. This results in

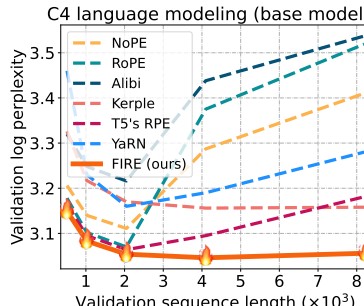

Figure 1: **Language modeling perplexity** on C4 with varying evaluation sequence lengths. Models are trained on length 2048.

a bounded input to the position encoding function for all input sequence lengths, leading to better generalization performance. As a result, with increasingly longer sequence lengths, the positional inputs will form progressively finer grids, interpolating the positional encoding function on $[0, 1]$.

Inspired by the existing methods, we incorporate the following two transformations into FIRE, which we find helpful to improve the model quality. i) To encourage locality bias in FIRE, we apply the popular $\log$ transformation (Raffel et al., 2019; Chi et al., 2022) to the relative distance before feeding it to the MLP, which amplifies the input differences for local tokens. ii) Next we modify progressive interpolation with a *learnable threshold* in the normalizer to yield exact distances for shorter contexts. Note that both these transformations do not limit the ability of the model to learn arbitrary biases. In fact we show that FIRE learns to pay more attention to far away contexts in some attention heads.

We conduct an extensive empirical study to demonstrate the effectiveness of FIRE for length generalization. We benchmark FIRE as well as other positional encoding approaches on a wide range of real-world language modeling (C4, arXiv, and Github), long text benchmark (SCROLLS), zero-shot long-context question answering (NarrativeQA), and natural language understanding benchmarks (GLUE/SuperGLUE). Our empirical results show the strong length generalization performance and long text modeling capability of FIRE. Our experiments on standard natural language understanding benchmarks show that FIRE is competitive on short sequence tasks as well. We further visualize the learned positional encoding of FIRE showing that it learns diverse patterns, beyond just locality bias.

The main contributions of our paper are summarized below:

- We propose FIRE, a new functional relative positional encoding method. Using progressive interpolation, FIRE is able to transform arbitrary input lengths into bounded domain, followed by a learned mapping.

- We theoretically prove that FIRE can represent popular position encodings such as T5's RPE, Alibi, and Kerple, thereby unifying a class of existing position encoding approaches.

- We empirically show strong length generalization behavior of FIRE, significantly improving over existing methods in zero-shot and finetuning settings on a wide range of datasets and benchmarks. For instance, it consistently delivers strongest performance on C4 language modeling across various sequence lengths, outperforming the best baseline by 2.28 perplexity points (Fig. 1). On SCROLLS long text benchmark, FIRE surpasses all the competing methods on average by over 1 point (Table 1).

- We present visualization of learned position embeddings of FIRE model showing that it can learn both local and anti-local position biases.

## 2 POSITIONAL ENCODINGS AND LENGTH GENERALIZATION

We are interested in building Transformer models with *length generalization* ability, i.e., we expect that the model can be trained on sequences of length $L_{\text{train}}$ and be directly applied to sequence length $L_{\text{test}}$ without performance degradation for $L_{\text{test}} > L_{\text{train}}$ (Press et al., 2022). Length generalization requires Transformers to generalize to unseen positions during training, and designing better position

encodings is an active line of research towards improving the length generalization (Chi et al., 2022; 2023; Kazemnejad et al., 2023; Chen et al., 2023). In this section, we review existing positional encoding approaches with an emphasis on their length generalization abilities. More discussions on related work can be found in Appendix D.

## 2.1 ABSOLUTE POSITIONAL ENCODING

The Transformer paper (Vaswani et al., 2017) proposes Absolute Positional Encoding (APE) to endow Transformers with positional information. In particular, a (learnable or fixed sinusoidal) real-valued embedding $e_i \in \mathbb{R}^d$ is assigned to each position $i$, leading to an Absolute Positional Encoding matrix $E = [e_1, \cdots, e_n]^\top$, which will be added to the input sequence. Though simple and straightforward, APE-based Transformers usually generalize poorly to longer sequences (Press et al., 2022).

## 2.2 RELATIVE POSITIONAL ENCODING

Relative Positional Encoding (RPE) is an increasingly popular way to encode positional information for Transformers. Shaw et al. (2018) are the first to introduce RPE to Transformers and their proposed method adds position encodings to the key (and optionally the value) in the attention layer, instead of the input. Raffel et al. (2019) simplify the vector representations of relative positions to scalars and use them as a bias term added to the pre-softmax attention logits. They further map any OOD sequence lengths to the same position, resulting in length generalization. This form of *additive RPE* has proven to be highly effective in many applications (Dai et al., 2019; Liu et al., 2021; Ying et al., 2021). Following this, multiple additive RPE methods have been proposed to improve both length generalization and efficiency, such as Alibi (Press et al., 2022), Kerple (Chi et al., 2022), and Sandwich (Chi et al., 2023).

**Additive RPE.**   For most of these additive RPE methods, the computation of the (pre-softmax) attention logits can be unified using the following formula:

$$A_{\mathrm{RPE}}(X) = XW_Q(XW_K)^\top + B, \tag{1}$$

where the bias matrix $B \in \mathbb{R}^{n \times n}$ is induced by the **position encoding function** $b : \mathbb{N}^{*2} \to \mathbb{R}$. Let the $(i, j)$-th entry of $B$ be $b(i, j)$. Different formulations and parameterizations of $b$ lead to different RPE variants. A few examples that support arbitary sequence length include:

- T5's RPE (Raffel et al., 2019): $b(i, j) = r_{\min\{i-j, K\}}$, where $K$ is a hyper-parameter and $\{r_i\}_{i=0}^K$ are learnable scalars.[3]
- Alibi (Press et al., 2022): $b(i, j) = -r|i - j|$, where $r > 0$ is a hyper-parameter.
- Kerple (Chi et al., 2022): $b(i, j) = -r_1 \log(1 + r_2|i-j|)$ (logarithmic variant) or $-r_1|i-j|^{r_2}$ (power variant), where $r_1, r_2 > 0$ are learnable scalars.
- Sandwich (Chi et al., 2023): $b(i, j) = r_1 \sum_{k=1}^{r_2} \cos\left((i - j)/10000^{\frac{k}{d'}}\right)$, where $r_1$ and $r_2$ are hyper-parameters.

The above methods can be applied to longer sequences than training, but they also have several limitations. T5's RPE uses the same attention bias for all query-key pairs with distance greater than $K$, lacking representational power to distinguish between different positions in long sequences. Furthermore, it relies on vector operations that are not accelerator-friendly, making its training and inference relatively slow (Press et al., 2022). Alibi, Kerple, and Sandwich significantly bias towards local attention, making it harder to attend to more distant query-key pairs (Chi et al., 2023). This property can prevent the model from capturing long-range dependencies and lead to performance degradation on some tasks. In the subsequent section, we will present our method to overcome these limitations.

**Rotary Positional Encoding.**   In addition to the aforementioned methods, there are also several non-additive RPE variants. Among them, the most popular one in large language models is Rotary Position Encoding (RoPE) (Su et al., 2021; Chowdhery et al., 2022; Touvron et al., 2023a). RoPE

---

[3]In practice, T5's RPE segments relative distances into distinct buckets with a logarithmic scale, each associated with a unique parameter. Refer to Appendix A.1 for further details.

rotates the query and key vectors with an angle proportional to their absolute positions before the dot product attention, which results in attention being a function of the relative distance between the tokens, capturing the relative positional information.

Press et al. (2022); Kazemnejad et al. (2023) find that RoPE-based language models have poor length generalization. To address this, Chen et al. (2023) propose RoPE with position interpolation, and show this allows better length generalization of these models. Such interpolation techniques ((Chen et al., 2023) for RoPE and (Dosovitskiy et al., 2021) for APE), usually requires 1) knowing the target sequence length *a priori*, which may not be feasible in practical generative applications, 2) finetuning the model at the new target sequence length, which can be challenging for larger scale models. In contrast, our proposed approach uses a progressive interpolation technique that does not require any prior information of the target sequence length. This property is appealing since the maximum sequence length can be hard to predict for auto-regressive language models. Further, our experiments show that the proposed approach does not require any additional finetuning to achieve strong zero-shot length generalization behavior.

## 2.3 NO POSITIONAL ENCODING

While encoder-only Transformer models (e.g., BERT (Devlin et al., 2019)) are permutation equivariant without positional encoding, Haviv et al. (2022) show that decoder-only Transformers with causal attention masks can learn positional information even without any explicit positional encoding. Recently, Kazemnejad et al. (2023) show that the no positional encoding (NoPE) model shows strong length generalization on small scale synthetic tasks.

## 3 METHOD

In this section, we formally introduce **FIRE** (**F**unctional **I**nterpolation for **R**elative Positional **E**ncoding), a new relative positional encoding approach for improving length generalization of Transformers.

### 3.1 FUNCTIONAL POSITION ENCODING WITH PROGRESSIVE INTERPOLATION

Our proposed approach FIRE uses a **learnable continuous function** to map input positions to biases. We implement the function using an MLP $f_\theta : \mathbb{R} \to \mathbb{R}$,[4] where $\theta$ denotes the MLP parameters. This avoids hard coding specific inductive biases and lets the position encoding be learnt jointly with the task at hand. A standard approach would be to feed the relative query-key distance as the input to the MLP. However this suffers from generalization issues when the inputs (the relative distances) are outside the training domain of the MLP.

We propose **Progressive Interpolation** to address this challenge. Instead of using the raw query-key relative distance as the input to the MLP, we normalize the distance by the query position index. Formally, we consider the following positional encoding function:

$$b(i, j) = f_\theta \left( \frac{i - j}{i} \right) \text{ where } f_\theta(x) = \boldsymbol{v}_3^\top \sigma(\boldsymbol{V}_2 \sigma(\boldsymbol{v}_1 x)), \; \theta = \{\boldsymbol{v}_1, \boldsymbol{V}_2, \boldsymbol{v}_3\}.^5 \tag{2}$$

Here $\sigma$ is the ReLU activation function; $i$ and $j$ denote the query and key positions respectively. Note that in causal attention, the relative distance satisfies $0 \le i - j < i$. Therefore, the normalized relative distance is constrained to be in $[0, 1]$ regardless of the sequence length. In particular, with increasingly longer sequence lengths, the positional inputs will form progressively finer grids, interpolating the positional encoding function on $[0, 1]$. Hence, this technique aligns inference domain with training domain for any sequence lengths, leading to better length generalization.

**Discussion on the choice of the normalizer.** FIRE uses the query position $i$ to normalize the relative distance and implement interpolation. For auto-regressive generation with causal attention,

---

[4]Here we focus on a single attention head. Generally, with $H$ heads, FIRE learns an MLP $f_\theta : \mathbb{R} \to \mathbb{R}^H$ and uses different attention biases for different heads.

[5]$\boldsymbol{v}_1, \boldsymbol{v}_3 \in \mathbb{R}^s$, $\boldsymbol{V}_2 \in \mathbb{R}^{s \times s}$, and $s$ denotes the hidden size. Bias terms are omitted for brevity.

the query position index $i$ corresponds to the length of *current* context. Another possible choice is to use some pre-defined max context length as the normalizer. In this case, the model will still suffer from unfamiliar (large) distances when the texts exceed the pre-defined max lengths, making such a choice suboptimal. Using the query position index as the normalizer avoids this issue.

## 3.2 ADDITIONAL TRANSFORMATIONS

Inspired by existing methods, we introduce two transformations on FIRE for further improvement. We note that these transformations do no limit the expressive power of FIRE to learn arbitrary biases.

**Amplifying the differences among local positions.** Existing works show that RPE attention biases change more rapidly for the local tokens than for the distant tokens (Khandelwal et al., 2018; Wang et al., 2021). Thus, it's appealing to consider some monotonically increasing transformation $\psi : \mathbb{N} \to \mathbb{R}_+$ with a monotonically decreasing slope (i.e., a concave function) to the relative distance, so that more modeling capacity can be allocated to learn RPE for local positions:

$$b(i, j) = f_\theta \left( \frac{\psi(i - j)}{\psi(i)} \right). \tag{3}$$

For example, in our experiments, we use $\psi : x \mapsto \log(cx + 1)$ where $c > 0$ is a learnable parameter. This transformation $\psi$ amplifies the differences among local positions. Note that, the $\log$ transformation is applied to both the relative distance and the normalizer. Thus, the MLP inputs are still constrained to $[0, 1]$ for any sequence lengths as long as $\psi$ is monotonically increasing.

**Thresholding the normalizer for better short sequence modeling.** While the progressive interpolation technique offers robust length generalization capabilities, our preliminary experiments indicate a marginal degradation in model performance for shorter sequences. We posit that it's because the actual relative distances are important in RPE of short sequences, while the normalization in progressive interpolation obfuscates this information. To address this, we introduce an adaptive thresholding mechanism, activating the progressive interpolation technique only for larger query position indices, i.e., long contexts. Specifically, we define a *learnable* threshold $L$ and only apply progressive interpolation when $i > L$. For short sequences with less than $L$ tokens, we use $\psi(L)$ to normalize the relative distance.

Based on the above, the positional encoding function of FIRE can be formulated as

$$b_{\text{FIRE}}(i, j) = f_\theta \left( \frac{\psi(i - j)}{\psi(\max\{L, i\})} \right), \tag{4}$$

where $\psi : \mathbb{N} \to \mathbb{R}_+$ is monotonically increasing and $L > 0$ is a learnable scalar. Our main experiments of FIRE are based Eq. (4) with $\psi : x \mapsto \log(cx + 1)$. We present experiments ablating these design choices in Appendix B.

## 3.3 EXPRESSIVENESS OF FIRE

In this subsection, we theoretically prove that FIRE can represent all the existing additive RPE approaches discussed in Sec. 2.2. This expressiveness allows FIRE to learn suitable position encoding functions from the data. We state this formally in the theorem below. The proof can be found in Appendix A.

**Theorem 3.1.** *Let $b_0$ be the positional encoding function of T5's RPE, Alibi, Kerple, or Sandwich as defined in Sec. 2.2. Consider FIRE function $b_{\text{FIRE}}(i, j)$ in Eq. (4). Given any sequence length $L_0 \in \mathbb{N}^*$, there exist some transformation $\psi$, threshold $L$, and MLP configuration (weights $\theta$ and activation function $\sigma$) such that $b_{\text{FIRE}}(i, j) = b_0(i, j)$ for any $0 < j \leq i \leq L_0$.*

**Remark.** We point out that our proof is constructive, and does not leverage the universal approximation property of MLP, i.e., the MLP does not need to be extremely wide or deep. In fact, FIRE is *parameter efficient* in the sense that it represents T5's RPE, Alibi, and Kerple with nearly the same number of parameters (up to a constant factor). Further, in all our experiments with FIRE, we show that a small MLP with a hidden size of 32 suffices for strong performances.

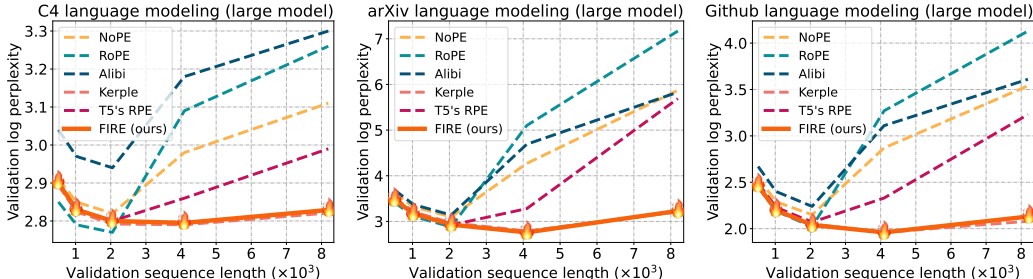

Figure 2: **Language modeling perplexity** with varying evaluation sequence lengths for large models trained on sequence length 2048.

# 4 EXPERIMENTS

In this section we present experimental results comparing our proposed unified relative encoding method FIRE with T5's RPE (Raffel et al., 2019), Alibi (Press et al., 2022), and Kerple (Chi et al., 2022), showing that the proposed approach significantly improves long context generalization while not sacrificing short context performance. We also include comparisons to other popular methods - Rotary Positional Encoding (RoPE) (Su et al., 2021) and no positional encoding (NoPE) (Kazemnejad et al., 2023). We use a hidden size of 32 for the MLPs in FIRE for all our experiments.

We consider language models trained on the C4 dataset (Raffel et al., 2019) with 2048 input length, with different positional encoding methods. We first compare the zero-shot perplexity values on inputs with different lengths (512 to 8192) from various datasets, comparing the long context generalization ability of different position encoding methods (Sec. 4.1). Later, we present finetuning results on both longer inputs of length 8192 on SCROLLS (Shaham et al., 2022) and shorter inputs of length 1024 on GLUE/SuperGLUE (Wang et al., 2019b;a) (Sec. 4.2 & 4.4).[6] In addition, we conduct experiments on zero-shot long-context question answering on NarrativeQA (Kočiský et al., 2018) with different context lengths from 512 to 32768 (Sec. 4.3). In Appendix B, we present some ablation experiments studying the design choices of FIRE. The complete experimental setup along with the hyper-parameters for each of the tasks and hardware details is provided in Appendix C.

## 4.1 LANGUAGE MODELING WITH LENGTH GENERALIZATION

Following Brown et al. (2020), we use the **causal LM** objective to pretrain decoder-only Transformers with different position encodings on C4 dataset (Raffel et al., 2019). We experiment with two model size settings, base (125M parameters) and large (350M parameters). The evaluation metrics are validation log perplexity on C4, arXiv, and Github (Raffel et al., 2019; Gao et al., 2020). We pretrain the models on sequence length to 2048, and evaluate their zero-shot perplexity on sequence lengths $\{512, 1024, 2048, 4096, 8192\}$. For base-sized models, we additionally compare our method with a concurrent work, YaRN (Peng et al., 2024), which improves length generalization of RoPE-based Transformer models.[7] Model and training configurations are detailed in Appendix C.1.

The results are shown in Fig. 1, 2, & 7. We first notice that FIRE consistently achieves lower perplexity across different model sizes, validation sequence lengths, and datasets. In comparison to existing approaches, the performance gain is particularly significant for validation sequences that are longer than training sequences (out-of-distribution sequence lengths), showing better length generalization behavior. For example, for base models trained on sequence length 2048 and evaluated on sequence length 8192, FIRE outperforms the best baseline method, Kerple, by 2.28 points (21.24 v.s. 23.52 perplexity). Methods such as RoPE achieve strong performance for in-distribution sequence lengths, but their performances quickly degrade with longer inputs. YaRN requires knowledge of target sequence length and further finetuning, but we can see from Fig. 1 & 7 that it underperforms FIRE on long sequences and sacrifices model quality on short sequences (e.g., length 512). Note that in all our experiments, perplexity is computed in a single forward pass for a given input, and we do not use any sliding window tricks during inference (Press et al., 2022).

---

[6]While finetuning is not the same as the zero-shot long-context generalization, it still measures the ability of the pre-trained model to adapt to longer inputs in the downstream applications.

[7]We note that YaRN needs additional tuning on long sequences. All the other methods in this subsection, including FIRE, are evaluated on long context *without any tuning*.

Table 1: **Experimental results on SCROLLS benchmark.** Abbreviations for dataset names: Qasper (Qas), ContractNLI (CNLI), QMSum (QMS), NarrativeQA (NQA), SummScreenFD (SumS), GovReport (GovR), and QuALITY (QuAL). We provide the evaluation metrics, the median sequence lengths in each dataset (Ainslie et al., 2023), and detailed results for base/large models. RoPE-PI refers to the RoPE interpolation (Chen et al., 2023). Best results are highlighted in **bold**.

| | QAS | CNLI | QMS | NQA | SumS | GovR | QuAL | Average |
|---|---|---|---|---|---|---|---|---|
| **Metric** | F1 | EM | Rgm | F1 | Rgm | Rgm | EM | |
| **Median length** | 5472 | 2148 | 14197 | 57829 | 9046 | 8841 | 7171 | |
| *Base models* | | | | | | | | |
| NoPE | 10.98 | 72.90 | 14.36 | 5.90 | 15.44 | 16.24 | 22.10 | 22.56 |
| RoPE | 10.44 | 71.75 | 14.90 | 8.71 | 14.40 | 15.72 | 6.71 | 20.38 |
| RoPE-PI | 15.41 | 71.94 | 13.12 | 9.21 | 15.77 | **16.86** | 20.33 | 23.23 |
| Alibi | 8.38 | 67.21 | 5.48 | 4.24 | 3.49 | 6.96 | 9.68 | 15.06 |
| Kerple | 11.67 | 75.99 | 14.39 | 9.24 | 15.73 | 16.42 | **25.36** | 24.11 |
| T5's RPE | 12.80 | 74.93 | **16.12** | 9.00 | 15.37 | 15.96 | 24.83 | 24.14 |
| FIRE (ours) | **16.24** | **82.93** | 14.58 | **9.55** | **15.87** | 16.31 | 24.02 | **25.64** |
| *Large models* | | | | | | | | |
| NoPE | 15.34 | 74.25 | 15.79 | 7.56 | 16.60 | 16.66 | 24.16 | 24.34 |
| RoPE | 11.01 | 79.94 | 15.13 | 9.40 | 15.84 | 15.50 | 9.92 | 22.39 |
| RoPE-PI | 17.02 | 84.28 | 14.05 | 10.14 | 16.72 | **17.03** | 23.01 | 26.04 |
| Alibi | 8.20 | 68.95 | 5.81 | 4.91 | 4.34 | 11.58 | 12.27 | 16.58 |
| Kerple | 18.93 | 77.24 | 15.09 | 9.97 | 17.14 | 16.85 | 24.83 | 25.72 |
| T5's RPE | 17.51 | 75.70 | **16.17** | 9.62 | 16.68 | 16.76 | 24.45 | 25.27 |
| FIRE (ours) | **19.47** | **85.15** | 15.10 | **10.27** | **17.27** | 16.83 | **25.26** | **27.05** |

## 4.2 FINETUNING ON LONG TEXT BENCHMARK

To further test the models' capability of learning and modeling long sequences, we conduct finetuning experiments on SCROLLS, a long text benchmark (Shaham et al., 2022) which contains 7 different datasets. We initialize the models with the C4 checkpoints pretrained on sequence length 2048, and finetune them on sequence length 8192 for each individual task. In addition to position encoding methods in Sec. 4.1, we also experiment with RoPE with positional interpolation (RoPE-PI) (Chen et al., 2023), which extends the context window of RoPE-based pretrained models given a downstream maximum sequence length. Following existing works by Shaham et al. (2022); Ainslie et al. (2023), we use three different evaluation metrics (Rgm, F1, and EM scores) for different datasets. We also compute the average score across different datasets as done in the SCROLLS benchmark. Detailed descriptions of the datasets and evaluation metrics are provided in Appendix C.2.

The results on SCROLLS benchmark are shown in Table 1. We first notice that FIRE attains the best average score, outperforming existing approaches by over 1.0 point on both model sizes. Even at the individual task level, FIRE achieves the best performances on 4/5 out of 7 tasks among the base/large models. RoPE-PI significantly improves RoPE as expected, but lags behind FIRE. One drawback though is that RoPE-PI requires the knowledge of maximum input sequence length beforehand, which is not always known in practice for decoder-only models.

## 4.3 ZERO-SHOT LENGTH GENERALIZATION ON NARRATIVEQA

We next evaluate the zero-shot length generalization capabilities of the finetuned models on the downstream NarrativeQA dataset. We use the NarrativeQA dataset (Kočiský et al., 2018) with different input context lengths to test the model's ability to leverage long context in zero-shot learning settings. We use the base-sized model checkpoints pretrained on C4 (sequence length 2048) and finetuned on NarrativeQA (sequence length 8192). We evaluate the models on context lengths {512, 2048, 4096, 8192, 16384, 24576, 32768} without any further tuning on the target context lengths. For RoPE with position interpolation (Chen et al., 2023), we consider two variants with max sequence lengths set to 8192 or 32768. We use unigram overlap (F1) as the evaluation metric.

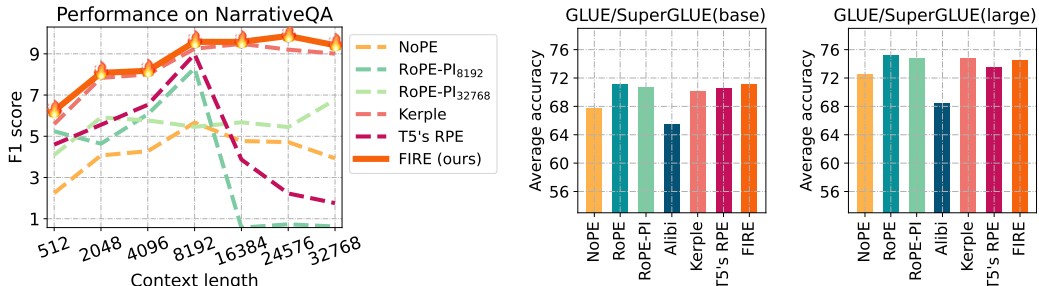

Figure 3: **Left: Comparisons on NarrativeQA with different context lengths.** "RoPE-PI$_{8192}$" and "RoPE-PI$_{32768}$" refers to RoPE interpolation with max sequence lengths 8192 and 32768 respectively. **Right: Results on GLUE and SuperGLUE benchmarks.** We report the average accuracy across all the tasks on these two benchmarks.

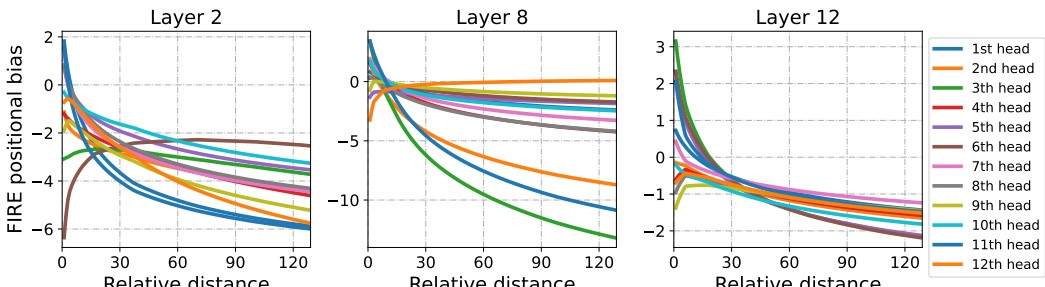

Figure 4: **Visualization of FIRE learned position biases** for the 128th query position with key positions between 1 and 128. We notice that FIRE learns both local and anti-local position patterns.

We compare FIRE with the most competitive baselines in the left panel of Fig. 3. Detailed results (including omitted baselines) can be found in Table 11. We notice that FIRE achieves top performances consistently across different sequence lengths. The plot also shows sensitivity of RoPE-PI to the max sequence length parameter in this zero-shot length generalization setting. Setting the max sequence length to a small value (8192) results in good performance until 8192, but with a steep drop for longer contexts. On the other hand, using a larger value for max sequence length (32768) gets rid of the steep drop for long contexts; but results in worse performance across all sequence lengths. In contrast, FIRE using progressive interpolation is able to generalize across all sequence lengths.

## 4.4 FINETUNING ON GLUE/SUPERGLUE

We next evaluate the C4 pre-trained models on GLUE/SuperGLUE benchmarks, to test these methods on shorter sequence lengths. We finetune on standard natural language understanding benchmarks, GLUE (Wang et al., 2019b) and SuperGLUE (Wang et al., 2019a), with shorter sequence lengths (1024) to evaluate the general quality of the models. We use the average accuracy/exact match across all the tasks as our main evaluation metric. Detailed experiment results can be found in Table 12.

The results are shown in the right of Fig. 3. Among the baseline approaches, NoPE and Alibi slightly lag behind, while RoPE, Kerple, and T5's RPE all achieve similarly good accuracies. FIRE is on par with these approaches, demonstrating good performance on GLUE and SuperGLUE tasks. These results show that although FIRE is designed to enhance length generalization of Transformers, it does not sacrifice the accuracy on downstream tasks with shorter sequence lengths.

## 4.5 VISUALIZATION OF FIRE

In this subsection we present visualization of learned position encoding biases from a FIRE model pretrained on C4. We plot the learned position encoding bias for the query token at the 128th position, for all the attention heads from selected layers in Fig. 4. We notice that, in different attention heads, FIRE learns both local and "anti-local" attention patterns that emphasize far away keys more, showing the advantage of functional approach, as opposed to a fixed local inductive bias (Press et al., 2022; Chi et al., 2022; 2023).

Table 2: **Comparing FIRE with/without positional encoding function sharing across layers.** FIRE and FIRE-S refer to models without and with sharing, respectively.

| | *C4 log perplexity with varying lengths* | | | | | | *GLUE & SuperGLUE* | |
| | **512** | **1024** | **2048** | **4096** | **8192** | | **Average accuracy** | |
|---|---|---|---|---|---|---|---|---|
| **FIRE** | 3.15 | 3.08 | 3.05 | 3.05 | 3.06 | | 71.14 | |
| **FIRE-S** | 3.22 | 3.14 | 3.10 | 3.09 | 3.10 | | 71.04 | |
| | *SCROLLS benchmark* | | | | | | | |
| | **Qas** | **CNLI** | **QMS** | **NQA** | **SumS** | **GovR** | **QuAL** | **Average** |
| **FIRE** | 16.24 | 82.93 | 14.58 | 9.55 | 15.87 | 16.31 | 24.02 | 25.64 |
| **FIRE-S** | 17.93 | 75.22 | 15.05 | 9.22 | 16.02 | 16.25 | 24.11 | 24.83 |

## 4.6 LAYERWISE SHARING

Another important factor beyond length generalization is computational cost of these approaches. Most of FIRE's computation is based on matrix multiplication, which is more accelerator-friendly than the vector operations used in T5's RPE. To further improve the computational efficiency of FIRE, we consider FIRE-S, a weight-sharing version which uses the same position encoding bias for all the layers. This way the position encoding bias only needs to be computed once, and the cost is amortized over all the layers. Note that sharing position encoding across layers is a common inductive bias in many existing methods (Su et al., 2021; Press et al., 2022; Luo et al., 2022).

We conduct experiments to evaluate FIRE-S (with layerwise sharing) on C4 language modeling, SCROLLS long text benchmark, and GLUE/SuperGLUE. We also measure the inference speed of different methods. Experimental details are provided in C.6.

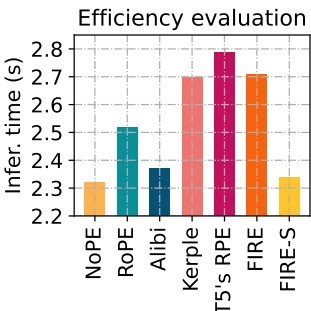

Figure 5: **Inference time comparisons** for different methods. The reported results are averaged over 10 runs for each method.

**Model quality.** Table 2 compares the accuracy of FIRE-S and the standard FIRE. The results show that sharing position encoding function across layers only leads to a slight performance degradation. FIRE-S still outperforms other baselines in the long sequence regime. For example, on C4 language modeling with sequence length 8192, it outperforms Kerple, the best baseline in Fig. 1 (3.10 v.s. 3.16 log perplexity). On SCROLLS, its average score outperforms all the strong baseline methods including T5's RPE, RoPE with positional interpolation, and Kerple.

**Inference speed.** Fig. 5 compares the model speed of FIRE/FIRE-S with baselines. We first notice that FIRE and FIRE-S are both faster than T5's RPE while achieving stronger performances. Moreover, FIRE-S significantly improve the efficiency of FIRE and is faster than all the baselines but NoPE (no positional encoding). In conclusion, the experiments show that FIRE-S demonstrates good speed-accuracy trade-off.

## 5 CONCLUSION

We propose a functional interpolation for relative position encoding (FIRE) to improve Transformer's ability to generalize to longer contexts, and present theoretical and empirical results showing its effectiveness. We prove that FIRE unifies many existing additive RPE methods, while being adaptive enough to learn diverse position encoding biases in long context settings. Empirical results show strong length generalization behavior pushing the paradigm of train short test long. Our work does suffer from some limitations. 1) We only study decoder models. 2) We do not analyze the role of other components of Transformer and other training components (data, optimizer) in length generalization. These questions are interesting directions for future exploration.

ACKNOWLEDGMENTS

This work is supported in part by the United States Department of Energy via the Brookhaven National Laboratory under Contract No. 384608.

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

## A OMITTED PROOF

In this section, we first provide a more general formulation of T5's positional encoding function as mentioned in Sec. 2.2. Then we provide the proof of Theorem 3.1.

### A.1 T5'S RPE WITH BUCKETING

In Sec. 2.2, we use a simplified description for T5's RPE. In practice, T5's RPE does not assign different position bias for *all* different relative positions. Instead, all possible relative distances are partitioned into several buckets, and the relative distances in one bucket share a (learnable) attention bias. Formally speaking, T5's RPE pre-defines $0 = s_0 < s_1 < \cdots < s_{k-1} < s_K$, and computes the attention bias as

$$b(i,j) = \begin{cases} r_k & s_k \leq i - j < s_{k+1}; \ k = 0, \cdots, K-1 \\ r_K & i - j \geq s_K \end{cases}. \tag{5}$$

It's easy to see that the formulation in Sec. 2.2 is a special case of Eq. (5) by setting $s_k = k$. In the official T5 implementation[8], the buckets are defined based on "log binning". With $K+1$ buckets and a pre-defined distance $L_1$, the attention bias is calculated as (assuming $K+1$ is even)

$$b(i,j) = \begin{cases} r_{i-j} & 0 \leq i - j < \frac{K+1}{2} \\ r_{\frac{K+1}{2} + \lfloor \frac{K+1}{2} \log\left(\frac{2(i-j)}{K+1}\right) / \log\left(\frac{2L_1}{K+1}\right) \rfloor} & \frac{K+1}{2} \leq i - j < L_1 \\ r_K & i - j \geq L_1 \end{cases}. \tag{6}$$

This is also a special case of Eq. (5).

In the proof of Theorem 3.1, we will be working on the most general formulation (Eq. (5)), so that the proof works for any specific instances.

### A.2 PROOF OF THEOREM 3.1

*Proof.* For each RPE variant (T5's RPE, Alibi, Kerple, and Sandwich), we provide constructions in which FIRE represent each of the target $b_0$ for $0 < j \leq i < L_0$.

**T5's RPE.** We consider the general T5's RPE formulation with bucketing in Eq. (5). The target positional encoding function can be rewritten as

$$b_0(i,j) = r_0 + \sum_{k=1}^{K} (r_k - r_{k-1}) \cdot \mathbb{1}_{\{i-j \geq s_k\}}. \tag{7}$$

Consider a two-layer MLP with activation $\sigma(x) = \mathbb{1}_{\{x \geq 0\}}$ and $K$ hidden neurons:

$$f_\theta(x) = \boldsymbol{v}_2^\top \sigma(\boldsymbol{v}_1 x + \boldsymbol{b}_1) + b_2. \tag{8}$$

Let $\boldsymbol{v}_1 = L_0 \boldsymbol{1}$ (where $\boldsymbol{1}$ denotes an all-one vector), $\boldsymbol{b}_1 = [-s_1, -s_2, \cdots, -s_K]^\top$, $\boldsymbol{v}_2 = [r_1 - r_0, r_2 - r_1, \cdots, r_K - r_{K-1}]^\top$, and $b_2 = r_0$.

In the positional encoding function of FIRE (Eq. (4)), we set the transform $\psi$ to be the identity mapping $x \mapsto x$ and the threshold $L$ to $L_0$.

---

[8] https://github.com/google-research/text-to-text-transfer-transformer.

Then for any $0 < j \leq i \leq L_0$,

$$b_{\text{FIRE}}(i, j) = f_\theta \left( \frac{i - j}{L_0} \right) \tag{9}$$

$$= \begin{bmatrix} r_1 - r_0 & r_2 - r_1 & \cdots & r_K - r_{K-1} \end{bmatrix} \sigma \left( \begin{bmatrix} i - j - s_1 \\ i - j - s_2 \\ \vdots \\ i - j - s_K \end{bmatrix} \right) + r_0 \tag{10}$$

$$= \begin{bmatrix} r_1 - r_0 & r_2 - r_1 & \cdots & r_K - r_{K-1} \end{bmatrix} \begin{bmatrix} \mathbb{1}_{\{i-j \geq s_1\}} \\ \mathbb{1}_{\{i-j \geq s_2\}} \\ \vdots \\ \mathbb{1}_{\{i-j \geq s_K\}} \end{bmatrix} + r_0 \tag{11}$$

$$= \sum_{k=1}^{K} (r_k - r_{k-1}) \cdot \mathbb{1}_{\{i-j \geq s_k\}} + r_0. \tag{12}$$

Thus, we have $b_{\text{FIRE}}(i, j) = b_0(i, j)$ for any $0 < j \leq i \leq L_0$.

**Alibi.** The target positional encoding function is $b_0(i, j) = -r(i - j)$ (note that we focus on the setting where $i \geq j$). Consider a one-layer MLP with identity activation and no bias term (which degrades to a linear mapping) $f_\theta(x) = v_1 x$, and let $v_1 = -r L_0$. In the positional encoding function of FIRE (Eq. (4)), we set the transform $\psi$ to be the identity mapping $x \mapsto x$ and the threshold $L$ to $L_0$. Then for any $0 < j \leq i \leq L_0$,

$$b_{\text{FIRE}}(i, j) = f_\theta \left( \frac{i - j}{L_0} \right) = -r(i - j) = b_0(i, j), \tag{13}$$

which concludes the proof.

**Kerple (logarithmic variant).** The target positional encoding function is $b_0(i, j) = -r_1 \log(1 + r_2(i - j))$ (note that we focus on the setting where $i \geq j$). Consider a one-layer MLP with identity activation and no bias term (which degrades to a linear mapping) $f_\theta(x) = v_1 x$. and let $v_1 = -r_1 \log(1 + r_2 L_0)$. In the positional encoding function of FIRE (Eq. (4)), we set the transform $\psi$ to be the log transform $x \mapsto \log(r_2 x + 1)$ and the threshold $L$ to $L_0$. Then for any $0 < j \leq i \leq L_0$,

$$b_{\text{FIRE}}(i, j) = f_\theta \left( \frac{\log(1 + r_2(i - j))}{\log(1 + r_2 L_0)} \right) = -r_1 \log(1 + r_2(i - j)) = b_0(i, j), \tag{14}$$

which concludes the proof.

**Kerple (power variant).** The target positional encoding function is $b_0(i, j) = -r_1 (i - j)^{r_2}$ (note that we focus on the setting where $i \geq j$). Consider a two-layer MLP with activation $\sigma(x) = x^{r_2}$, one hidden neuron, and no bias term: $f_\theta(x) = v_2 (v_1 x)^{r_2}$. Let $v_1 = \sqrt[r_2]{r_1} L_0$ and $v_2 = -1$. In the positional encoding function of FIRE (Eq. (4)), we set the transform $\psi$ to be the identity mapping $x \mapsto x$ and the threshold $L$ to $L_0$. Then for any $0 < j \leq i \leq L_0$,

$$b_{\text{FIRE}}(i, j) = f_\theta \left( \frac{i - j}{L_0} \right) = -(\sqrt[r_2]{r_1}(i - j))^{r_2} = -r_1 (i - j)^{r_2} = b_0(i, j), \tag{15}$$

which concludes the proof.

**Sandwich.** The target positional encoding function is

$$p_0(i, j) = c \sum_{k=1}^{d'} \cos \left( (i - j)/10000^{\frac{k}{d'}} \right). \tag{16}$$

Consider a two-layer MLP with $\cos$ activation, $d'$ hidden neurons, and no bias term:

$$f_\theta(x) = \boldsymbol{v}_2^\top \cos(\boldsymbol{v}_1 x). \tag{17}$$

Let $\boldsymbol{v}_1 = \left[ L_0/10000^{\frac{1}{d'}}, L_0/10000^{\frac{2}{d'}}, \cdots, L_0/10000^1 \right]^\top$ and $v_2 = c\mathbf{1}$. In the positional encoding function of FIRE (Eq. (4)), we set the transform $\psi$ to be the identity mapping $x \mapsto x$ and the threshold $L$ to $L_0$. Then for any $0 < j \leq i \leq L_0$,

$$b_{\mathrm{FIRE}}(i, j) = f_\theta\left(\frac{i-j}{L_0}\right) \tag{18}$$

$$= \begin{bmatrix} c & c & \cdots & c \end{bmatrix} \begin{bmatrix} \cos\left((i-j)/10000^{\frac{1}{d'}}\right) \\ \cos\left((i-j)/10000^{\frac{2}{d'}}\right) \\ \vdots \\ \cos\left((i-j)/10000^1\right) \end{bmatrix} \tag{19}$$

$$= c\sum_{k=1}^{d'} \cos\left((i-j)/10000^{\frac{k}{d'}}\right). \tag{20}$$

Thus, we have $b_{\mathrm{FIRE}}(i, j) = b_0(i, j)$ for any $0 < j \leq i \leq L_0$. □

## B  ABLATION STUDY

The positional encoding function of FIRE can be viewed as a composition of a *position transformation* and a *function approximator* $b(i, j) = f_\theta(g(i-j, i))$. The position transformation $g$ takes the relative distance $i - j$ and the query position $i$ as the input and produces a "normalized" distance. For example, in Eq. (4), the position transformation $g : (i - j, i) \mapsto \psi(i-j)/\psi(\max\{i, L\})$. Different choices of $\psi$ leads different position transformation $g$. The function approximator $f_\theta$ should be in an expressive function class parametrized by $\theta$, which transforms the normalized distances into attention biases. For example, we use a two-hidden-layer MLP with 32 neurons in each hidden layer and ReLU activation by default, as discussed in Appendix C.1.

In this section we ablate our design choices for both the position transformation and the function approximator. We also conduct ablation experiments to test the length generalization performances on different training sequence lengths. All the ablation experiments are based on base-sized models.

### B.1  THE LOG TRANSFORM AND THRESHOLDING IN POSITION TRANSFORMATIONS

In Sec. 3.2, we propose two modifications, the $\log$ transformation and thresholding operation, as additional transformations to the relative distance. We conduct experiments to ablate these design choices and demonstrate their effectiveness. We experiment with base-sized models and compare FIRE variants with or without the additional transformations in Sec. 3.2. Specifically, we consider three variants with the following positional encoding functions:

Without $\log$ transform/thresholding: $b_1(i, j) = f_\theta\left(\frac{i-j}{i}\right).$ (21)

With $\log$ transform but without thresholding: $b_2(i, j) = f_\theta\left(\frac{\log(c(i-j)+1)}{\log(ci+1)}\right).$ (22)

With $\log$ transform and thresholding: $b_3(i, j) = f_\theta\left(\frac{\log(c(i-j)+1)}{\log(c\max\{L, i\}+1)}\right).$ (23)

For all the three variants (Eq. (21-23)), $f_\theta$ is parameterized as a two-hidden-layer MLP with 32 neurons in each hidden layer and ReLU activation to ensure a fair comparison. Eq. (23) is the standard FIRE positional encoding function used in Sec. 4. We experiment on C4 language modeling and GLUE/SuperGLUE benchmark using the settings and evaluation metrics described in Appendix C. The experimental results are shown in Table 3. From the language modeling results, we can see that both the log transformation and the thresholding operation improve the language modeling quality for all the lengths, and the standard FIRE positional encoding function in Eq. (23) is the best variant. In particular, the $\log$ transformation largely improve the performance on long sequences,

indicating that amplifying the differences among local positions helps in the long sequence regimes. We further study the effectiveness of the thresholding operation on GLUE/SuperGLUE benchmark which contains relatively short sequences. The results show that the thresholding operation leads to 0.72 point performance gain on average GLUE/SuperGLUE accuracy, verifying its effectiveness on improving short sequence modeling.

Table 3: **Ablation study on the position transformation.** We compare FIRE variants with or without the additional transformations in Sec. 3.2. For log transform, ✗ indicates $\psi(x) = x$, i.e., no log transform; while ✓ indicates $\psi(x) = \log(cx + 1)$, i.e., applying log transform for the relative distance. For thresholding, ✗ indicates using $\psi(i)$ to normalize the relative distance, i.e., thresholding operation; while ✓ indicates $\psi(\max\{i, L\})$ to normalize the relative distance with $L$ being a learnable threshold.

| Method | | | C4 log perplexity with varying lengths | | | | |
|---|---|---|---|---|---|---|---|
| **Log transform** | **Thresholding** | **Formula** | **512** | **1024** | **2048** | **4096** | **8192** |
| ✗ | ✗ | Eq. (21) | 3.194 | 3.128 | 3.099 | 3.216 | 3.334 |
| ✓ | ✗ | Eq. (22) | 3.161 | 3.093 | 3.062 | 3.057 | 3.085 |
| ✓ | ✓ | Eq. (23) | 3.149 | 3.083 | 3.054 | 3.046 | 3.056 |

| Method | | | GLUE/SuperGLUE |
|---|---|---|---|
| **Log transform** | **Thresholding** | **Formula** | **Average accuracy** |
| ✗ | ✗ | Eq. (21) | 69.06 |
| ✓ | ✗ | Eq. (22) | 70.42 |
| ✓ | ✓ | Eq. (23) | 71.14 |

**Additional discussions on the thresholding operation.** We note that even FIRE *without* thresholding outperforms all the baselines (including RoPE, T5's RPE, etc) on all the sequence lengths on C4 language modeling. Detailed comparisons are in Table 4.

In all the experiments presented in the paper, the threshold $L$ of FIRE in Eq. (23) is a *learnable* parameter. For the base-sized model pretrained on sequence length 2048, the learned parameter $L$ is between 1200 to 1600 across different layers. Setting $L$ to a fixed value is also a viable option. In our preliminary exploration, FIRE with either fixed or learnable $L$ outperforms all the baselines, while the learnable variant leads to better performances. The fixed variant introduces one more hyper-parameter and may require more tuning. Thus, FIRE uses learnable threshold $L$ as the default choice.

Table 4: **Comparing FIRE variants with baselines.** We present additional comparisons between existing methods and FIRE variants with or without thresholding.

| | C4 log perplexity with varying lengths | | | | |
|---|---|---|---|---|---|
| Method | **512** | **1024** | **2048** | **4096** | **8192** |
| NoPE | 3.206 | 3.14 | 3.111 | 3.287 | 3.410 |
| RoPE | 3.178 | 3.102 | 3.070 | 3.375 | 3.519 |
| Alibi | 3.320 | 3.248 | 3.216 | 3.438 | 3.537 |
| Kerple | 3.326 | 3.217 | 3.170 | 3.156 | 3.158 |
| T5's RPE | 3.164 | 3.095 | 3.064 | 3.095 | 3.181 |
| FIRE without thresholding (Eq. (22)) | 3.161 | 3.093 | 3.062 | 3.057 | 3.085 |
| FIRE (Eq. (23)) | 3.149 | 3.083 | 3.054 | 3.046 | 3.056 |

## B.2 EFFECTS OF THE FUNCTION APPROXIMATOR CAPACITY ON THE PERFORMANCES

We experimentally study the impact of the function approximator ($f_\theta$) capacity on the model performance. We compare a linear layer, a one-hidden-layer MLP, and a two-hidden-layer MLP. The MLPs both have 32 neurons in the hidden layers and use ReLU (Nair & Hinton, 2010) activation function. Two-hidden-layer MLP is the defualt choice for FIRE in Sec. 4. We experiment on C4 language

modeling and evaluate the models on varying sequence lengths using the settings and evaluation metrics described in Appendix C.1 and present the experiment result in 5. The result shows that a linear layer is not experssive enough and leads to suboptimal performance on C4 language modeling. Introducing non-linearty and parametrizing $f_\theta$ as a one/two-hidden-layer MLP leads to much better results. In particular, using a one-hidden-layer MLP has largely improve the overall performances especially in the long sequence regimes. For example, it outperforms a linear $f_\theta$ by 0.24 point log perplexity on sequence length 8192. Moreover, using an MLP with larger capacity (two hidden layers v.s. one hidden layer) can further brings performance gains. That being said, the MLP is still very tiny (with only 32 hidden neurons) and we believe it's the non-linearty that helps.

Table 5: **Ablation study on the capacity of the function approximator ($f_\theta$).** We compare FIRE variants with different activation functions in MLP.

| Parametrization of $f_\theta$ | C4 log perplexity with varying lengths | | | | |
|---|---|---|---|---|---|
| | **512** | **1024** | **2048** | **4096** | **8192** |
| Linear | 3.21 | 3.14 | 3.11 | 3.20 | 3.32 |
| One-hidden-layer MLP (32 hidden neurons) | 3.17 | 3.10 | 3.07 | 3.06 | 3.08 |
| Two-hidden-layer MLP (32 hidden neurons) | 3.15 | 3.08 | 3.05 | 3.05 | 3.06 |

### B.3 CHOICE OF THE MLP ACTIVATION FUNCTION

We study the impact of the MLP activation function on the model performance. We experiment on C4 language modeling and evaluate the models on varying sequence lengths using the settings and evaluation metrics described in Appendix C.1. We compare ReLU (Nair & Hinton, 2010) and GeLU (Hendrycks & Gimpel, 2016) activation functions and present the experiment result in 6. The result shows that the model performance is not sensitive to the choice of activation function in the length generalization setting, while ReLU works better on normal sequence lengths. Thus, we use ReLU as our default activation function.

Table 6: **Ablation study on the MLP activation.** We compare FIRE variants with different activation functions in MLP.

| | C4 log perplexity with varying lengths | | | | |
|---|---|---|---|---|---|
| | **512** | **1024** | **2048** | **4096** | **8192** |
| ReLU | 3.15 | 3.08 | 3.05 | 3.05 | 3.06 |
| GeLU | 3.36 | 3.26 | 3.06 | 3.05 | 3.06 |

### B.4 CHOICE OF FINAL ACTIVATION OF MLP OUTPUT

In our main experiments, we focus on MLPs of the form $f_\theta(x) = \boldsymbol{v}_\ell^\top \sigma(\cdots \sigma(\boldsymbol{v}_1 x))$ where $\sigma$ is the activation function. In this implementation, the MLP ends with a linear layer and no activation function is applied to the MLP final output. A slightly different choice is to consider $\tilde{f}_\theta(x) = \sigma(\boldsymbol{v}_\ell^\top \sigma(\cdots \sigma(\boldsymbol{v}_1 x)))$ where a final activation is applied to the MLP output. We compare these two choices by experimenting on C4 language modeling and evaluating the models on varying sequence lengths. We use one-hidden-layer MLP with 32 hidden neurons and the ReLU (Nair & Hinton, 2010) activation function in both model variants. The results are presented in Table 7. We find that MLP without final activation leads to better performances on long sequences and use it as our default choice.

### B.5 FIRE IS STILL STRONG WHEN TRAINED ON SEQUENCE LENGTH 512

In most of our pretraining experiments, the training sequence length is set to 2048 (see Appendix C.1). In this experiment we train models with different positional encodings on C4 with training sequence length 512 to confirm that the overall performance trends are not sensitive to the pretraining

Table 7: **Ablation study on final activation of MLP output.** We compare FIRE variants using MLP with/without final activation to its output.

| | C4 log perplexity with varying lengths | | | | |
| | 512 | 1024 | 2048 | 4096 | 8192 |
| --- | --- | --- | --- | --- | --- |
| With final activation | 3.16 | 3.10 | 3.07 | 3.09 | 3.19 |
| Without final activation | 3.17 | 3.10 | 3.07 | 3.06 | 3.08 |

sequence length. Other experimental settings are te same as those in Appendix C.1. We evaluate the models on varying sequence lengths and report the log perplexity in Fig. 6. It's clear that FIRE still achieves the strongest overall performance compared with all the other baselines. The results in Fig. 1 & 6 demonstrate that FIRE can robustly deliver higher modeling quality regardless of the training sequence lengths.

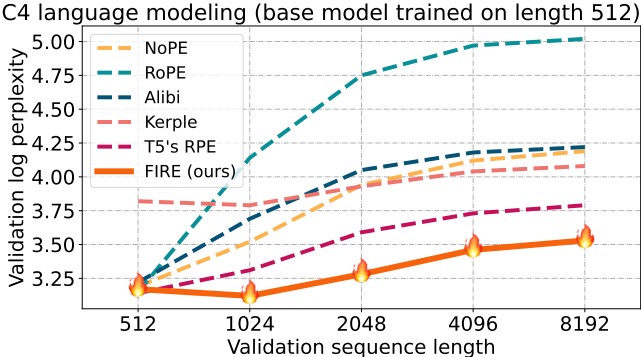

Figure 6: **Language modeling perplexity** evaluated on varying sequence lengths on C4 validation set. The plots are base-sized models with training sequence length 512.

## C   EXPERIMENT SETTINGS & ADDITIONAL RESULTS

### C.1   LANGUAGE MODELING WITH LENGTH GENERALIZATION

**Model configurations.**   In this experiment, we train decoder-only Transformer language models with different positional encoding variants while keeping all the other configurations the same. For T5's RPE, we follow Raffel et al. (2019) and use 64 position bucket for each attention head. For Alibi, we follow Raffel et al. (2019) to set the hyperparameters in the positional encoding function in each attention head. For our FIRE method, we use the positional encoding function defined in Eq. (4). In Eq. (4), we let $\psi(x) = \log(cx + 1)$ where $c$ is a learnable parameter; $f_\theta$ is parametrized as a two-hidden-layer MLP with 32 neurons in each hidden layer and $\mathrm{ReLU}$ activation.

We experiment with two model size settings, base (125M parameters) and large (350M parameters). The model configurations follow (Brown et al., 2020) and are presented in Table 8.

**Training recipe.**   Following Brown et al. (2020), we use the **causal LM** objective to pretrain decoder-only Transformers with different position encodings. We use the C4 dataset (Raffel et al., 2019) as the pretraining corpora. We set pretraining sequence lengths to 2048, and evaluate the zero-shot perplexity on sequence lengths $\{512, 1024, 2048, 4096, 8192\}$. We truncate documents with length greater than 2048 to multiple sequences of length 2048 during training; similar trucation is done to construct the validation sets of different sequence lengths. Our training recipe follows (Brown et al., 2020) and is presented in Table 9.

**Additional results.**   We evaluate language modeling log perplexity with varying lengths on C4, arXiv, and Github datasets (Raffel et al., 2019; Gao et al., 2020) for both base and large models. The

Table 8: **Model configurations** for language model pretraining.

|  | Small model | Large model |
|---|---|---|
| Training sequence length | 2048 | 2048 |
| Number of layers | 12 | 24 |
| Attention heads | 12 | 16 |
| Hidden layer size | 768 | 768 |
| Head dimensions | 64 | 64 |
| FFN activation | GeLU | GeLU |
| Number of parameters | 125M | 350M |

results of base models on C4 are presented in Fig. 1. The results of large models on all the three datasets are presented in Fig. 2. In Fig. 7, we additionally present the results of base models on arXiv and Github. All the results show similar trends and FIRE consistently demonstrate strong length generalization behavior.

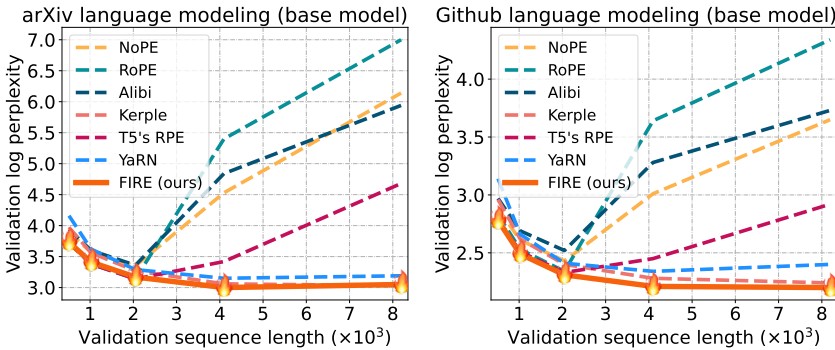

Figure 7: **Language modeling perplexity** evaluated on varying sequence lengths on **arXiv (left)** and **Github (right)** validation set. The plots are base-sized models with training sequence length 2048.

Table 9: **Training recipe** for language model pretraining.

|  | Small model | Large model |
|---|---|---|
| Training sequence length | 2048 | 2048 |
| Batch size | 256 | 256 |
| Numer of iterations | 600k | 600k |
| Dropout prob. | 0.0 | 0.0 |
| Attention dropout prob. | 0.0 | 0.0 |
| Optimizer | AdamW | AdamW |
| Learning rate | $6e-4$ | $3e-4$ |
| Hardware (TPUv4 chips) | 128 | 256 |

## C.2 FINETUNING ON LONG TEXT BENCHMARK

**Datasets and evaluation metrics.** We use SCROLLS long text benchmark (Shaham et al., 2022) to further test the models' capability of learning and modeling long sequences. SCROLLS benchmark includes question-answering datasets - Qasper, NarrativeQA, and QuALITY ; natural language inference datasets - ContractNLI; and summarization datasets - QMSum, SummScreenFD, and GovReport. Following existing works Shaham et al. (2022); Ainslie et al. (2023), three different evaluation metrics are used for different datasets: Rgm score (the geometric mean of ROUGE-1,2,L) for GovReport, SummScreenFD, and QMSum, unigram overlap (F1) for Qasper and NarrativeQA, and exact match (EM) for ContractNLI and QuALITY. We also compute the average score across different datasets as done in the SCROLLS benchmark.

**Model and training configurations.**    We finetune the checkpoints pretrained on C4, so the model configurations are the same as those in Table 8. We use the same set of hyperparameters for all the models and all the tasks, and report the best results on the validation set. Table 10 presents our finetuning configurations.

Table 10: **Finetuning configurations** for SCROLLS benchmark.

| | |
|---|---|
| Batch size | 128 |
| Numer of iterations | 25k |
| Dropout prob. | 0.1 |
| Attention dropout prob. | 0.1 |
| Optimizer | AdamW |
| Learning rate | $1e-5$ |
| Hardware (TPUv4 chips) | 128 |

## C.3    ZERO-SHOT LENGTH GENERALIZATION ON NARRATIVEQA

**Datasets and evaluation metrics.**    We use the NarrativeQA dataset (Kočiskỳ et al., 2018) with different input context lengths to test the model's ability to leverage long context in zero-shot learning settings. We use the base-sized model checkpoints pretrained on C4 (sequence length 2048) and finetuned on NarrativeQA (sequence length 8192). We evaluate the models on context lengths $\{512, 2048, 4096, 8192, 16384, 24576, 32768\}$ and use unigram overlap (F1) as the evaluation metric.

**Detailed results.**    We provide detailed performances of all the tested models in Table 11. The result shows that FIRE is consistently outperforming all the baselines across all different context lengths.

Table 11: **Detailed performance comparisons on NarrativeQA with varying context lengths.** "RoPE-PI$_{L_0}$" refers to RoPE interpolation with max sequence lengths $L_0$. Best performances are highlighted in **bold**.

| Context length | 512 | 2048 | 4096 | 8192 | 16384 | 24576 | 32768 | Average |
|---|---|---|---|---|---|---|---|---|
| NoPE | 2.245 | 4.070 | 4.277 | 5.661 | 4.770 | 4.716 | 3.930 | 4.238 |
| RoPE | 1.546 | 1.482 | 2.060 | 8.737 | 1.071 | 0.190 | 0.132 | 2.174 |
| RoPE-PI$_{8192}$ | 5.241 | 4.639 | 6.070 | 8.301 | 0.565 | 0.728 | 0.623 | 3.738 |
| RoPE-PI$_{32768}$ | 4.092 | 5.912 | 5.769 | 5.459 | 5.677 | 5.446 | 6.767 | 5.589 |
| Alibi | 4.036 | 4.339 | 4.190 | 4.251 | 4.144 | 4.086 | 3.899 | 4.135 |
| Kerple | 5.590 | 7.832 | 8.001 | 9.249 | 9.483 | 9.204 | 9.010 | 8.338 |
| T5's RPE | 4.595 | 5.557 | 6.528 | 8.983 | 3.872 | 2.226 | 1.757 | 4.788 |
| FIRE (ours) | **6.232** | **8.076** | **8.178** | **9.581** | **9.581** | **9.868** | **9.417** | **8.705** |

## C.4    FINETUNING ON GLUE/SUPERGLUE

**Datasets, evaluation metrics, and configurations.**    GLUE and SuperGLUE are widely-used benchmarks to evaluation the natrual language understanding capability of neural language models (Wang et al., 2019b;a). We finetune the models on a mixture of the tasks in GLUE and SuperGLUE for simplicity. We evaluate the model on each task separately. We use the macro average accuracy/exact match across all the tasks as our main evaluation metric. Table 12 presents our finetuning configurations.

**Detailed results.**    For reference, we present detailed results for all the models on each individual dataset in Table 13. In general, FIRE achieves decent performances. Thus, FIRE's strong performances on long sequences does not come at the price of sacrificing model quality on short sequences and standard tasks.

Table 12: **Finetuning configurations** for GLUE/SuperGLUE benchmark.

| | |
|---|---|
| Batch size | 256 |
| Numer of iterations | 25k |
| Dropout prob. | 0.1 |
| Attention dropout prob. | 0.1 |
| Optimizer | AdamW |
| Learning rate | $1e-5$ |
| Hardware (TPUv2 chips) | 32 |

Table 13: **Detailed performances on GLUE and SuperGLUE tasks.** The evaluation metrics are EM (exact match) for Multirc & Record; and accuracy for the remaining tasks.

*Base models*

| | Boolq | Cb | Cola | Copa | Mnli | Mrpc | Qnli | Qqp |
|---|---|---|---|---|---|---|---|---|
| NoPE | 72.51 | 73.21 | 69.42 | 67.00 | 79.72 | 75.98 | 84.70 | 88.72 |
| RoPE | 75.78 | 80.36 | 74.78 | 60.00 | 83.11 | 79.17 | 87.70 | 90.03 |
| RoPE-PI | 75.72 | 80.36 | 72.87 | 64.00 | 82.87 | 80.64 | 86.89 | 89.93 |
| Alibi | 69.76 | 76.79 | 69.32 | 58.00 | 78.02 | 76.72 | 83.97 | 88.14 |
| Kerple | 77.31 | 82.14 | 74.11 | 61.00 | 82.69 | 80.64 | 87.66 | 90.22 |
| T5's RPE | 76.30 | 83.93 | 71.33 | 61.00 | 82.10 | 81.37 | 87.61 | 89.87 |
| FIRE (ours) | 76.76 | 83.93 | 73.63 | 59.00 | 83.01 | 80.39 | 87.83 | 89.97 |

| | Rte | Sst2 | Wic | Wnli | Multirc | Record | Wsc |
|---|---|---|---|---|---|---|---|
| NoPE | 71.84 | 91.17 | 58.78 | 63.38 | 16.89 | 35.50 | 67.31 |
| RoPE | 73.65 | 92.89 | 66.93 | 61.97 | 23.19 | 46.57 | 71.15 |
| RoPE-PI | 71.48 | 91.51 | 65.05 | 60.56 | 22.46 | 45.96 | 70.19 |
| Alibi | 68.23 | 88.76 | 57.05 | 61.97 | 12.70 | 29.34 | 63.46 |
| Kerple | 69.68 | 92.43 | 64.89 | 53.52 | 22.56 | 47.74 | 66.35 |
| T5's RPE | 73.65 | 92.20 | 63.79 | 60.56 | 20.57 | 45.71 | 69.23 |
| FIRE (ours) | 75.81 | 92.66 | 64.58 | 60.56 | 25.81 | 46.89 | 66.35 |

*Large models*

| | Boolq | Cb | Cola | Copa | Mnli | Mrpc | Qnli | Qqp |
|---|---|---|---|---|---|---|---|---|
| NoPE | 79.27 | 83.93 | 78.24 | 61.00 | 84.39 | 79.90 | 89.79 | 90.74 |
| RoPE | 79.66 | 91.07 | 80.54 | 63.00 | 85.67 | 81.86 | 90.87 | 91.04 |
| RoPE-PI | 79.45 | 92.86 | 80.54 | 63.00 | 85.31 | 81.62 | 90.52 | 91.05 |
| Alibi | 74.77 | 80.36 | 71.05 | 58.00 | 81.72 | 79.41 | 86.18 | 89.75 |
| Kerple | 80.70 | 92.86 | 79.29 | 65.00 | 85.63 | 80.88 | 90.56 | 90.86 |
| T5's RPE | 79.88 | 87.50 | 78.33 | 65.00 | 84.80 | 83.58 | 89.77 | 90.71 |
| FIRE (ours) | 79.60 | 85.71 | 79.10 | 65.00 | 84.93 | 81.13 | 90.37 | 90.84 |

| | Rte | Sst2 | Wic | Wnli | Multirc | Record | Wsc |
|---|---|---|---|---|---|---|---|
| NoPE | 77.26 | 93.69 | 62.70 | 59.16 | 26.65 | 51.18 | 70.19 |
| RoPE | 79.42 | 94.38 | 69.59 | 60.56 | 30.64 | 58.23 | 72.12 |
| RoPE-PI | 79.06 | 94.61 | 70.69 | 56.34 | 31.17 | 56.69 | 68.27 |
| Alibi | 72.56 | 91.97 | 60.35 | 50.70 | 22.77 | 40.79 | 66.35 |
| Kerple | 79.06 | 94.61 | 67.24 | 53.52 | 31.17 | 58.55 | 71.15 |
| T5's RPE | 79.78 | 92.89 | 64.58 | 54.93 | 29.80 | 52.54 | 69.23 |
| FIRE | 80.87 | 93.92 | 67.71 | 59.16 | 31.90 | 54.67 | 72.12 |
| Kerple | 79.06 | 94.61 | 67.24 | 53.52 | 31.17 | 58.55 | 71.15 |
| T5's RPE | 79.78 | 92.89 | 64.58 | 54.93 | 29.80 | 52.54 | 69.23 |
| FIRE (ours) | 80.87 | 93.92 | 67.71 | 59.16 | 31.90 | 54.67 | 72.12 |

## C.5   Visualization

We present another visualization of learned FIRE biases for query at position 8192 in Figure 8.

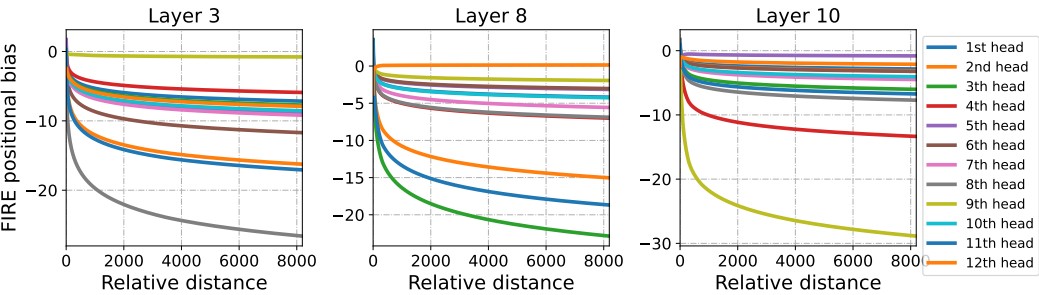

Figure 8: **Visualization of FIRE learned position biases** for the 8192nd query position with key positions between 1 and 8192. We notice that FIRE models learn both local and anti-local position patterns.

## C.6   Efficiency and FIRE-Shared

For FIRE-S (FIRE with layerwise sharing), we experiment with the base-sized model (125M parameters), and keep all the configurations and training recipes the same as those in previous subsections. The models are pretrained on C4 with sequence length 2048. The finetuning sequence lengths are 8192/1024 for SCROLLS and GLUE/SuperGLUE, respectively.

For the inference time evaluation, we test the forward time of base-sized model with different positional encodings on sequence length 2048. We measure the forward time on 4 TPUv2 chips for all the models, and report the average result over 10 runs.

## D   Related Works

In the main body of the paper, we cover the most relevant works to our paper (Sec. 2). In this section, we provide more discussions on related works.

**Length generalization.**   Many existing works show the length generalization failure of standard Transformer models (Press et al., 2022; Anil et al., 2022; Deletang et al., 2023; Liu et al., 2024). Recently, there have been growing interests in long-context applications such as multi-step reasoning (Wei et al., 2022; Dziri et al., 2023; Zhao et al., 2023) and document/book understanding (Kočiskỳ et al., 2018; Ke et al., 2022; Guo et al., 2022; Ainslie et al., 2023; Liu et al., 2023). Designing length-generalizable Transformers is appealing for these applications. Dubois et al. (2020); Chowdhury & Caragea (2023) introduce location attention for length generalization on synthetic tasks. Bueno et al. (2022) show that generating step-by-step rationales and using marker tokens as positional guides helps length generalization. Studying positional encoding approaches for length generalization is a main direction in this line of research. Press et al. (2022); Chi et al. (2022; 2023) propose new relative positional encoding methods which emphasize recency bias and improve language modeling on longer sequences. Chu et al. (2023) propose Conditional Positional Encodings to enhance Vision Transformer length generalization. The most relevant to our work is a concurrent paper by Chen et al. (2023). It propose Position Interpolation (PI) for Rotary Positional Encoding (RoPE), which extends the context window of RoPE-based pretrained models given a downstream max sequence length. However, this requires additional finetuning on longer sequence data, albeit for much fewer steps than original training. By contrast, our proposed FIRE does not require a pre-defined max sequence length, and can be directly applied to length generalization setting without tuning. We provide extensive experimental comparisons in Sec. 4. More recently, Zhou et al. (2024) show that standard Transformers can generalize to a sequence length that is 2.5× the training input length on integer addition using FIRE (and other techniques (Ruoss et al., 2023; Zhou et al., 2023)).

**Positional encoding in Transformers.** Positional encoding is a critical component of Transformers. Vaswani et al. (2017) propose sinusoidal Absolute Positional Encoding (APE) to encode positional information in the sequential input. Shaw et al. (2018) are the first to propose Relative Positional Encoding (RPE) for Transformers, and many follow-up works explore different RPE strategies (Dai et al., 2019; Raffel et al., 2019). There are also many works that study positional encoding from different perspectives, including the disentanglement of positional and content information (Kitaev & Klein, 2018; Ke et al., 2021), the representational power of attention modules and Transformers (Cordonnier et al., 2019; Chen et al., 2021; Li et al., 2021; Luo et al., 2022), computational efficiency (Su et al., 2021; Liutkus et al., 2021; Luo et al., 2021; Choromanski et al., 2023), and length generalization (Press et al., 2022; Chi et al., 2022; 2023; Kazemnejad et al., 2023). Our work is based on a unified formulation of existing additive relative positional encoding approaches, and proposes new RPE variant aimed at improving length generalization.

**Interpolation techniques in deep learning.** Interpolation techniques are successfully applied to many deep learning applications, especially in computer vision. Long et al. (2015) employ bilinear interpolation in up-sampling layers of convolutional neural networks for dense visual prediction. Dong et al. (2015); Johnson et al. (2016) employ bicubic interpolation for image super-resolution. Radford et al. (2015) probe generative models by interpolation in the latent space. Zhang et al. (2018); Han et al. (2022) use interpolating between pairs of examples and their labels as an data augmentation method. Recently, Dosovitskiy et al. (2021) propose to perform 2D interpolation of the pre-trained APE for Vision Transformer to apply the model to higher resolution images. In contrast, our interpretation is applied in the relative position encoding functions. Besides, we are focused on causal attention setting where "global" information such as the total sequence length is unknown, while Dosovitskiy et al. (2021) work on encoder-only Transformers with fixed input lengths.

## E    IMPLEMENTATION

In this section, we present the implementation of our proposed FIRE module in `PyTorch` (Paszke et al., 2019).

```
1  import torch
2  import torch.nn as nn
3
4  class FIRE(nn.Module):
5    def __init__(self, num_heads=12, mlp_width=32, init_c=0.1,
6                 init_L=512., eps=1e-6):
7      """
8      FIRE attention bias module.
9
10     Args:
11       num_heads: number of attention heads.
12       mlp_width: Width of MLP.
13       init_c: initial value of log transformation parameter
14       init_L: initial value of thresholding parameter
15       eps: small constant for numerical stability
16     """
17     super(FIRE, self).__init__()
18
19     # Define the MLP layers
20     self.mlp = nn.Sequential(
21       nn.Linear(1, mlp_width),
22       nn.ReLU(),
23       nn.Linear(mlp_width, num_heads)
24     )
25
26     # Initialize c (log transformation parameter)
27     self.c = nn.Parameter(torch.tensor(init_c))
28
```

```
29      # Initialize L (threshold)
30      self.init_L = nn.Parameter(torch.tensor(init_L),
31                               requires_grad=False)
32      # Learn a multiplier to L
33      self.L_multiplier = nn.Parameter(torch.tensor(1.0))
34
35      self.eps = eps
36
37  def forward(self, x: torch.Tensor):
38      """
39      Compute FIRE attention bias.
40
41      Args:
42        x: input sequence,
43            shape [bsz, num_heads, seq_len, hidden_dim]
44
45      Returns:
46        attention bias,
47        shape [1, num_heads, seq_len, seq_len]
48      """
49      seq_length = x.size(2)
50      positions = torch.arange(seq_length,
51                              dtype=torch.float,
52                              device=x.device)
53      rel_distance = positions[:, None] - positions[None, :]
54
55      # Thresholding the normalizer
56      threshold = torch.abs(self.L_multiplier * self.init_L)
57      pos_normalizer = torch.max(positions, threshold)
58      pos_normalizer = pos_normalizer[:, None]
59
60      # Amplifying differences among local positions
61      # with log transform
62      rel_distance = torch.log(
63        torch.abs(self.c * rel_distance) + 1
64      )
65      pos_normalizer = torch.log(
66        torch.abs(self.c * pos_normalizer) + 1
67      ) + self.eps
68
69      # Progressive interpolation
70      normalized_distance = rel_distance / pos_normalizer
71      fire_bias = self.mlp(normalized_distance.unsqueeze(-1))
72      fire_bias = fire_bias.unsqueeze(0).permute(0, 3, 1, 2)
73      return fire_bias
```

