# OpenReview forum: "Functional Interpolation for Relative Positions improves Long Context Transformers"
_ICLR.cc/2024/Conference — ICLR 2024 poster_

### Official Review · Reviewer_hDCR · 2023-11-01

**Soundness:** 3 good
**Presentation:** 3 good
**Contribution:** 3 good
**Rating:** 6
**Confidence:** 4

**Summary:**

Preventing performance degradation of Transformers on inputs longer than the training sequence lengths has posed a significant challenge in expanding the context length of these models. While the Transformer architecture inherently has no limitations on the input sequence lengths it can handle, the choice of position encoding during training can limit their performance on longer inputs. To address this, the author propose a novel approach called Functional Interpolation with Relative Encoding (FIRE), which aims to enhance Transformer's generalization to longer contexts. The author provide theoretical evidence that FIRE can effectively represent popular relative position encodings like T5's RPE, Alibi, and Kerple. Furthermore, The author empirically demonstrate that the FIRE models exhibit improved generalization to longer contexts in both zero-shot language modeling and long text benchmarks.

**Strengths:**

The author introduces a new relative position encoding and demonstrates its superior extrapolation capabilities.

**Weaknesses:**

Although the performance of the model presented in this paper is impressive, I believe there are several shortcomings:

1. Heavy reliance on handcrafted input features: The authors attempt to use a neural network to learn a relative position encoding. However, the input features of this network still heavily rely on manual design. Although some ablation experiments are conducted in section B.1, I believe the baseline should be a simpler approach like b(i, j) = f(i - j). For example, [1], [2] has achieved good results by learning relative positional relationships using a similar approach.

2. Trade-off between efficiency and performance: Compared to Alibi and RoPE, the learnable FIRE raises the question of how much slower it makes the training process for language modeling. However, this aspect is not mentioned in the paper. Therefore, the authors need to evaluate and discuss the training speed.

3. Lack of certain baselines: For instance, the paper fails to compare its approach with NTK-RoPE[3], YaRN[4], and NTK-ALiBi[5]. These methods have also made advancements in relative position encoding. The absence of comparison with these baselines is a drawback.


Citations:
[1] Zhen Qin, Xiaodong Han, Weixuan Sun, Bowen He, Dong Li, Dongxu Li, Yuchao Dai, Lingpeng Kong, and Yiran Zhong. Toeplitz neural network for sequence modeling. In The Eleventh International Conference on Learning Representations (ICLR), 2023

[2] Daniel Y. Fu, Elliot L. Epstein, Eric Nguyen, Armin W. Thomas, Michael Zhang, Tri Dao, Atri Rudra, and Christopher Ré. Simple hardware-efficient long convolutions for sequence modeling. CoRR, abs/2302.06646, 2023

[3] https://www.reddit.com/r/LocalLLaMA/comments/14lz7j5/ntkaware_scaled_rope_allows_llama_models_to_have/

[4] https://github.com/jquesnelle/yarn

[5] https://github.com/keezen/ntk_alibi/blob/main/readme_en.md

**Questions:**

The same as weaknesses part.

---

> ### Author Response · Authors · 2023-11-16
> **Official Comment by Authors (1/2)**
>
> Thank you for your careful review! Your comments point out very meaningful avenues for further improvement of our work. We respond to your concerns as below.
>
> **Regarding Weakness 1.** Thanks for the reference! We note that [1] and [2] propose new deep learning architectures for sequence modeling, while our work focuses on the Transformer variants. Besides, The idea of "simple" baseline $b(i,j)=f(i-j)$ is exactly T5's RPE, which has been extensively compared in our experiments. This "simple" baseline, along with many existing methods, are mainly evaluated using perplexity-based measures in previous works. Our systematic experiments show that they suffer significant performance drops with longer contexts across diverse evaluation tasks. Those methods typically have hard-coded inductive biases (with fixed RPE patterns or the same encoding for distant tokens), which prevent them from being universally effective. Our method addresses these drawbacks by using a learnable approach with interpolation to truly enable length generalization.
>
> To reiterate, progressive interpolation is proposed to map relative distances to numbers in $[0,1]$ and avoid out-of-distribution issues during inferences. The $\psi$ transform and thresholding technique further enhance modeling local positions and short sequences. We emphasize that these transformations do not limit the expressive power of FIRE to learn arbitrary biases, which is theoretically shown in Theorem 3.1. Moreover, we carefully ablate these design choices through experiments in Appendix B. Thus, we believe the design of FIRE is both practically effective and theoretically principled.
>
> **Regarding Weakness 2.**  Please note that the performance gap for long context settings between FIRE and baselines is so large ($>2$ perplexity points and $>1$ average SCROLLS score), shifting speed-accuracy pareto curve significantly. In fact, even base-sized FIRE is better than or on par with many large-sized baselines on SCROLLS. Further, in Sec. 4.6, we show that FIRE can be accelerated by layerwise sharing the positional encoding (FIRE-S) and provides the inference speed evaluations. We also follow your suggestion to additionally compare the speed and accuracy of RoPE, Alibi, and FIRE below:
>
> |               | C4 log perplexity (length 8192) | SCROLLS average score | Training speed (steps/sec) |
> |:-------------:|:-------------------------------:|:---------------------:|:--------------------------:|
> |  RoPE (base)  |               3.52              |         20.83         |           12.80            |
> |  RoPE (large) |               3.26              |         22.39         |            4.79            |
> |  Alibi (base) |               3.54              |         15.06         |           12.20            |
> | Alibi (large) |               3.30              |         16.58         |            4.58            |
> | FIRE-S (base) |               3.10              |         24.83         |           10.44            |
>
> It's clear that our method obtains much stronger performance with decent training speed. We will include a full version of the table in the final draft.

---

> ### Author Response · Authors · 2023-11-16
> **Official Comment by Authors (2/2)**
>
> **Regarding Weakness 3.** Thanks for the references! We would like to first highlight that all these 3 references appeared only after May 28, 2023, with the YaRN paper being uploaded on August 31st, just a month before the ICLR deadline. Hence we did not have comparisons in our submission.
>
> According to ICLR policy, _authors are not required to compare their own work to papers published on or after May 28, 2023_. As all the references have appeared after that and have not been peer-reviewed (we are not even able to find paper drafts for [3] and [5]), they should be considered _concurrent_ to our work.
>
> Among these references, YaRN [4] seems to be the only one with a formal preprint paper and a complete codebase. We follow your suggestion and additionally compare FIRE with YaRN. The results are shown below:
>
> -  **Language modeling on C4**
>
> | Sequence length |  512 | 1024 | 2048 | 4096 | 8192 |
> |:---------------:|:----:|:----:|:----:|:----:|:----:|
> |       YaRN      | 3.46 | 3.23 | 3.16 | 3.19 | 3.28 |
> |       FIRE      | 3.16 | 3.09 | 3.06 | 3.05 | 3.06 |
>
> -  **Language modeling on arXiv**
>
> | Sequence length |  512 | 1024 | 2048 | 4096 | 8192 |
> |:---------------:|:----:|:----:|:----:|:----:|:----:|
> |       YaRN      | 4.16 | 3.62 | 3.29 | 3.15 | 3.19 |
> |       FIRE      | 3.73 | 3.41 | 3.17 | 3.00 | 3.05 |
>
> -  **Language modeling on github**
>
> | Sequence length |  512 | 1024 | 2048 | 4096 | 8192 |
> |:---------------:|:----:|:----:|:----:|:----:|:----:|
> |       YaRN      | 3.14 | 2.66 | 2.41 | 2.34 | 2.40 |
> |       FIRE      | 2.78 | 2.49 | 2.31 | 2.21 | 2.20 |
>
>
> -  **SCROLLS benchmark**
>
> |      |  QAS  |  CNLI |  QMS  |  NQA |  SumS |  GovR |  QuAL | Average |
> |:----:|:-----:|:-----:|:-----:|:----:|:-----:|:-----:|:-----:|:-------:|
> | YaRN | 14.64 | 73.10 | 13.83 | 9.46 | 16.20 | 16.74 | 23.11 |  23.87  |
> | FIRE | 16.24 | 82.93 | 14.58 | 9.55 | 15.87 | 16.31 | 24.02 |  25.64  |
>
> The performance of YaRN is worse than FIRE on most datasets and sequence lengths, though it improves over RoPE PI. In [4], YaRN is claimed to be better than NTK-RoPE, so FIRE should also be better than NTK-RoPE. Besides, our experiments show that Alibi-based models consistently underperform RoPE/FIRE-based models. Therefore, we believe NTK-Alibi may be less promising to compare with.
>
> [1] Qin, Zhen, et al. "Toeplitz Neural Network for Sequence Modeling." arXiv preprint arXiv:2305.04749 (2023).
>
> [2] Fu, Daniel Y., et al. "Simple hardware-efficient long convolutions for sequence modeling." arXiv preprint arXiv:2302.06646 (2023).
>
> [3] https://www.reddit.com/r/LocalLLaMA/comments/14lz7j5/ntkaware_scaled_rope_allows_llama_models_to_have/
>
> [4] Peng, Bowen, et al. "Yarn: Efficient context window extension of large language models." arXiv preprint arXiv:2309.00071 (2023).
>
> [5] https://github.com/keezen/ntk_alibi/blob/main/readme_en.md
>
> We sincerely hope that our responses address your concerns and you reevaluate our work based on the responses. We are also willing to discuss with you if you have any further questions.

---

### Official Review · Reviewer_WBMp · 2023-11-06

**Soundness:** 3 good
**Presentation:** 3 good
**Contribution:** 3 good
**Rating:** 6
**Confidence:** 4

**Summary:**

In this work, the authors propose FIRE: functional interpolation for relative positional encoding. The idea is to first project relative position to a real number between 0 and 1, and this projected number is subsequently fed to a MLP to generate positional bias. FIRE enables language models to train on a short sequence length, and then generalize to longer sequence length during testing. Extensive experiments are conducted to demonstrate the effectiveness of the proposed method.

**Strengths:**

* The proposed method is very intuitive and easy to understand. The presentation is very clear. The main motivation is that many existing positional encoding methods cannot extrapolate to unseen sequence length, limiting their practicality. The idea of projecting relative positions to a real number between $[0,1]$ is very natural.

* The authors conduct extensive experiments to demonstrate the effectiveness of the proposed method. Models pre-trained with different positional encoding approaches are evaluated under different settings: fine-tuning on long and short context taks, and zero-shot generalization to unseen sequence length.

**Weaknesses:**

* Some of the design choices are ad-hoc, especially the thresholding parameter for the normalizer. From Table 3, it seems model performance considerably degrades on short sequence length without this thresholding parameter. I’m wondering about the model performance of using Eq. 21 on GLUE/SuperGLUE.

* Also regarding the thresholding parameter, it is mentioned that the parameter $L$ is learnable. Could the authors demonstrate what the learned parameter $L$ looks like after training? Also, will model performance significantly change if we set $L$ to a fixed value?

* Another weakness is that FIRE is not better than existing methods when the sequence length is short. Even with the thresholding parameter and the slower inference speed, it seems FIRE is not better than RoPE on GLUE/SuperGLUE.

**Questions:**

See above

---

> ### Author Response · Authors · 2023-11-16
>
> Thank you for supporting our work! We address your concerns as below.
>
> **Regarding the thresholding technique.** Thank you for your careful reading! We would like to first note that even FIRE _without_ thresholding _outperforms all the baselines_ (including RoPE, T5's RPE, etc) on all the sequence lengths. The thresholding technique is proposed to further enhance FIRE which turns out to be highly effective as demonstrated in our experiments. For your convenience, we reiterate the performance comparison in the table below. These results are in Sec. 4.1 & Appendix B.1 in the paper.
>
> |Method \ Test sequence length| 512 | 1024  | 2048  | 4096  | 8192  |
> |---------------------------|-------|-------|-------|-------|-------|
> | NoPE                      | 3.206 | 3.14  | 3.111 | 3.287 | 3.41  |
> | RoPE                      | 3.178 | 3.102 | 3.07  | 3.375 | 3.519 |
> | Alibi                     | 3.32  | 3.248 | 3.216 | 3.438 | 3.537 |
> | Kerple                    | 3.326 | 3.217 | 3.17  | 3.156 | 3.158 |
> | T5's RPE                  | 3.164 | 3.095 | 3.064 | 3.095 | 3.181 |
> | FIRE without thresholding | 3.161 | 3.093 | 3.062 | 3.057 | 3.085 |
> | FIRE                      | 3.149 | 3.083 | 3.054 | 3.046 | 3.056 |
>
>
> We follow your suggestion and additionally conduct GLUE/SuperGLUE finetuning for the variant based on Eq. (21). The averaged accuracy is 69.06. We will include this result in Table 3 for a more complete comparison.
>
> **Regarding the learned thresholding parameter $L$.** This is a good question. For the base-sized model with pretraining sequence length 2048, the learned parameter $L$ is between 1200 to 1600 across different layers.
>
> We agree that setting $L$ to a fixed value is another viable option. In our preliminary exploration, FIRE with a fixed/learnable $L$ achieved 24.65/25.64 average SCROLLS scores respectively. Both versions outperform all the baselines, while the learnable variant leads to better performances. Moreover, the fixed variant introduces one more hyper-parameter and may require more tuning. Thus, FIRE uses learnable $L$ as the default choice.
>
> **Regarding the short sequence performance.** We agree with the reviewer that FIRE does not lead to performance gains on GLUE/SuperGLUE, but is on par with existing approaches. We emphasize that the primary focus of FIRE is length generalization on long sequences, and our experiments on GLUE/SuperGLUE only serve as an additional sanity check of performances on shorter sequences. RoPE shows good performances on GLUE/SuperGLUE (normal sequence length), but it significantly underperforms FIRE on long sequences (Sec. 4.1-4.3). Besides, Sec. 4.6 shows that FIRE-S (with layer-wise sharing) is faster than RoPE, and the GLUE/SuperGLUE performance is comparable to RoPE (FIRE-S 71.04 v.s. RoPE 71.15). We believe it is a merit of our work to conduct more comprehensive evaluations rather than cherry-picking experimental settings.
>
> We sincerely hope that our responses address your concerns and you reevaluate our work based on the responses. We are also willing to discuss with you if you have any further questions.

---

### Official Review · Reviewer_VXMT · 2023-11-06

**Soundness:** 3 good
**Presentation:** 4 excellent
**Contribution:** 3 good
**Rating:** 8
**Confidence:** 4

**Summary:**

The paper proposes to use a small neural network that computes the positional bias used in self attention. The authors show that such a positional encoding is a generalization of several popular relative position encodings. Importantly, when combined with progressive interpolation (normalizing by the current position), they showcase superior performance in length generalization when using the proposed positional encoding. The authors perform extensive experiments on zero-shot length generalization as well as fine-tuning with longer sequences and show that the proposed method outperforms a host of other positional encodings.

**Strengths:**

The paper is very well written. The related work is clearly presented as is the proposed method.

Given the simplicity of the contribution, extensive experiments are required to ensure the universality of the proposed positional encoding. The authors perform experiments at two model scales, evaluating zero-shot generalization as well as generalization with fine-tuning. Moreover the authors provide extensive ablation studies for all components of the method.

**Weaknesses:**

It is not very clear why ALiBi performs so poorly while both in the Kerple paper and ALiBi paper, ALiBi shows great zero-shot length generalization.

Additionally, a simple baseline is missing, namely interpolating a learnable positional bias using $\frac{\psi(i - j)}{\psi(i)}$. This would show whether the nugget of the method is the progressive interpolation or the neural network or both. From the visualization of the learned positional biases the functions learned by the neural network are, as expected, not too complicated.

**Questions:**

Asked in the weaknesses section.

---

> ### Author Response · Authors · 2023-11-16
>
> Thank you for supporting our work! We address your concerns below.
>
> **Regarding the performance of Alibi.** We would like first note that our evaluation is much more comprehensive (as the reviewer noted earlier) including zero shot and finetuning on downstream tasks, whereas [1,2] mainly rely on language modeling performance for their evaluation. Second, for the language modeling task, we use standard evaluation to compute perplexity where we do a single forward pass on the entire sentence, whereas Alibi uses sliding window to evaluate perplexity. These differences could account for the discrepancies observed in Alibi's performance between our study and those reported in [1,2].
>
> [1] Press, Ofir, Noah Smith, and Mike Lewis. "Train Short, Test Long: Attention with Linear Biases Enables Input Length Extrapolation." International Conference on Learning Representations. 2021.
>
> [2] Chi, Ta-Chung, et al. "Kerple: Kernelized relative positional embedding for length extrapolation." Advances in Neural Information Processing Systems 35 (2022): 8386-8399.
>
> **Regarding the simple baseline.** Thanks for this suggestion. In the ablation study of our submission, we have tested a FIRE variant $f_{\theta}\left(\frac{\psi(i-j)}{\psi(\max\\{i, L\\})}\right)$ where $f_{\theta}$ is a linear function instead of a neural network. The results, presented in Table 4, show that using a linear $f_{\theta}$ leads to a 0.26-point drop in log perplexity on sequence length 8192, though is still better than competing baseline methods. This finding indicates using a neural-network-parametrization for $f_{\theta}$ is also important for the final performance. Besides, although the visualization shows the learned positional encoding is not very complex, the patterns still exhibit some non-linearity which may require a neural network to model.
>
> We sincerely hope that our responses address your concerns. Let us know if you have any further questions.

---

### Author Response · Authors · 2023-11-21
**Gentle Reminder to Review Authors' Responses for Paper 6081**

Dear Reviewers,

Thank you again for your time and effort in reviewing our paper. If you could spare a moment to review our response and share any further comments, it would be greatly appreciated!

Warm regards,

Authors of Paper 6081

---

### Meta-Review · Area_Chair_CoAW · 2023-12-04

**Metareview:**

This paper proposes a relative position encoding mechanism for improving a Transformer's generalization to contexts longer than those seen in training. It involves a simple learned function whose input is the relative position.

On the plus side, this approach is simple and leads to clear empirical gains across multiple benchmarks. On the minus side, it is unclear whether generalization beyond the context length seen during training is allowing it to **use** longer context information, or simply making sure the model does not degrade horribly (Figure 2 would suggest the latter, while the performance on SCROLLS would suggest the former). It is moreover unclear whether there is an interaction effect between this parameterization of the positional bias, and continual finetuning on longer context data, which has been popular recently.

**Justification For Why Not Higher Score:**

While there is some generalization to longer contexts than seen during, there is still performance degradation beyond 2x length generalization. Also unclear how important this issue is given that finetuning on longer-context data works well and is simple.

**Justification For Why Not Lower Score:**

Clear empirical gains across multiple benchmarks combined with the simplicity of the method.

---

### Decision · Program_Chairs · 2024-01-16

Accept (poster)